# TiAl6V4 Alloy Surface Modifications and Their Impact on Biofilm Development of *S. aureus* and *S. epidermidis*

**DOI:** 10.3390/jfb12020036

**Published:** 2021-05-18

**Authors:** Astrid H. Paulitsch-Fuchs, Lukas Wolrab, Nicole Eck, Nigel P. Dyer, Benjamin Bödendorfer, Birgit Lohberger

**Affiliations:** 1Biomedical Sciences, University of Applied Sciences Carinthia, 9020 Klagenfurt, Austria; A.Paulitsch-Fuchs@fh-kaernten.at (A.H.P.-F.); l.wolrab@cuas.at (L.W.); beno.boedendorfer@gmx.at (B.B.); 2Diagnostic and Research Institute of Hygiene, Microbiology and Environmental Medicine, Medical University of Graz, 8010 Graz, Austria; 3Department of Orthopaedics and Trauma, Medical University of Graz, 8036 Graz, Austria; nicole.eck@medunigraz.at; 4Bioinformatics Research Technology Platform, University of Warwick, Coventry CV4 7AL, UK; nigel.dyer@warwick.ac.uk

**Keywords:** *Staphylococcus aureus*, *Staphylococcus epidermidis*, biofilms, titanium alloys, prosthetic infections

## Abstract

One of the most serious complications following joint replacement surgeries are periprosthetic infections (PIs) arising from the adhesion of bacteria to the artificial joint. Various types of titanium–aluminum–vanadium (TiAl6V4) alloy surface modifications (coatings with silver (Ag), titanium nitride (TiN), pure titanium (cpTi), combinations of cpTi and hydroxyapatite (HA), combinations of cpTi and tricalcium phosphate (TCP), and a rough-blasted surface of TiAl6V4) have been investigated to assess their effects on biofilm development. Biofilms were grown, collected, and analyzed after 48 h to measure their protein and glucose content and the cell viability. Biofilm-associated genes were also monitored after 48 h of development. There was a distinct difference in the development of staphylococcal biofilms on the surfaces of the different types of alloy. According to the findings of this study, the base alloy TiAl6V4 and the TiN-coated surface are the most promising materials for biofilm reduction. Rough surfaces are most favorable when it comes to bacterial infections because they allow an easy attachment of pathogenic organisms. Of all rough surfaces tested, rough-blasted TiAl6V4 was the most favorable as an implantation material; all the other rough surfaces showed more distinct signs of inducing the development of biofilms which displayed higher protein and polysaccharide contents. These results are supported by RT-qPCR measurements of biofilm associated genes for *Staphylococcus aureus* (*icaA*, *icaC*, *fnbA*, *fnbB*, *clfB*, *atl*) and *Staphylococcus epidermidis* (*atle*, *aap*).

## 1. Introduction

Periprosthetic infections following the surgical treatment of bone defects are some of the most serious complications in orthopedics, and are caused mainly by *Staphylococcus aureus* (*S. aureus*) and *Staphylococcus epidermidis* (*S. epidermidis*) [1]. The initial step in such infections is the adhesion of bacteria to the foreign surface, resulting in a biofilm, which then leads to an inflammation of the periprosthetic tissue [2]. Whether an infection occurs is determined by a variety of factors such as passive adsorption through physicochemical interactions between bacteria and foreign surfaces or adhesins, and varies significantly between individual bacterial species [3,4,5]. The standard procedure for the treatment of periprosthetic infections is the two-sided replacement of the prosthesis [1,6,7]. However, pronounced loss of bone substance often occurs during surgery, and it is not always possible to preserve the body part in question when a replacement of the prothesis is necessary [8]. Furthermore, these infections are associated with high morbidity and mortality and their treatment is lengthy as well as costly, amounting up to EUR 50,000 per patient [9]. Titanium and titanium alloys such as TiAl6V4 have a wide range of medical applications. Due to its high biocompatibility, strength, resistance to corrosion, and similarity to trabecular bone tissue, TiAl6V4 is suitable as a raw material in orthopedics for endoprostheses. However, metallic prostheses such as those made of TiAl6V4 can lead to bone atrophy as well as sequential implant loosening, and they do not offer active protection against periprosthetic joint infections [10]. The combination of these above-mentioned issues, together with steadily increasing life expectancies and the high costs of implant replacement, create a need of improvements to the materials used [9].

Numerous physical and chemical surface modifications can and have been applied to metallic biomaterials in general [11], specifically to titanium-based materials [12], and most recently to the application of nano-modifications [13]. This has led to the invention of many new types of titanium-based orthopedic materials with not only a higher resistance to corrosion and better tribological properties [14,15], but also improved mechanical properties [16,17,18,19]. In addition, biocompatibility [19] and resistance to bacterial infections [12,20,21,22] can be positively influenced by those newly developed materials. As mentioned above, *S. aureus* and *S. epidermidis* are especially prone to cause device-related infections, and they are used in many biofilm-related studies in the field of orthopedics [6,21,22,23,24,25,26,27].

For this study, a pure TiAl6V4 base [5,27,28] was modified by adding silver (Ag), titanium nitride (TiN), pure titanium (cpTi), cpTi with hydroxyapatite (HA), and cpTi with tricalcium phosphate (TCP) surface coatings, as well as rough-blasting (rb) the surface of TiAl6V4. Using an in vitro model, the influence of these newly developed surface modifications on *S. aureus* and *S. epidermidis* biofilm development has been studied using protein and glucose quantification assays, live/dead cell distinction flow cytometric measurements, as well as the RT-qPCR of genes associated with biofilm development.

## 2. Materials and Methods

**TiAl6V4 surface modifications:** The tested alloy materials were manufactured by Implantcast GmbH (Buxtehude, Germany). According to the ISO 5832-3 [29] chemical characterization, TiAl6V4 has the following elemental composition (in %): Al 5.5–6.75, V 3.5–4.5, O ≤ 0.2, N ≤ 0.05, H ≤ 0.015, Fe ≤ 0.3, C ≤ 0.08, and the remainder consisting of titanium. The mechanical parameters were tested with the ISO 6892-1 [30] method and are given according to the ISO 5832-3 specifications: adhesive tensile strength ≥ 860 MPa, yield strength ≥ 860 MPa, and an elongation at break for rod shape ≥10 and for a sheet or tape ≥ 8. All material samples were in the form of 14 mm diameter discs with a thickness of 1 mm. The silver-coated TiAl6V4 discs had an electrolytically deposited, sandblasted metal layer with a silver coating thickness of 15 ± 5 µm and an adhesive strength compliant to the thermal shock experiment DIN EN ISO 4521:2009-01 [31] attachment C-C.6. Silver coatings are known to show antibacterial effects, caused by the intervention in the respiratory chain which has been confirmed for both Gram-positive and Gram-negative bacteria [32]. However, this cytotoxicity can also affect bone and tissue cells, and the release of silver ions might have a toxic effect on the patients (high serum and blood Ag levels) as well [33,34]. The discs were coated with titanium nitride (TiN) using cathodic arc deposition, a technique frequently used to synthesize extremely hard films to protect the surfaces of materials. The TiN-target used for deposition had the following elemental composition (in %): Fe ≤ 0.2, O ≤ 0.25, C ≤ 0.1, N ≤ 0.03, H ≤ 0.0125, with the remainder consisting of titanium. The coating had a thickness of 5.5 ± 1.5 µm, an adhesive tensile strength of ≥22 MPa, and a layer roughness of Ra ≤ 0.05 µm. This ceramic surface coating is known to possess anti-allergic and wear-reducing properties as well as heightened biocompatibility [35]. The titanium coating was applied using a vacuum plasma spray (VPS) of commercially pure titanium resulting in a 300 ± 50 µm coating thickness and an adhesive tensile strength of greater than 22 MPa and an average roughness of Ra 50 ± 15 µm. The maximum percentage of impurities in cpTi was as follows (in %): N ≤ 0.05, C ≤ 0.1, H ≤ 0.05, Fe ≤ 0.5, O ≤ 0.4 and the remainder consisting of titanium. In vitro studies have shown that due to the surface structure and chemistry, especially the increase in surface energy at the grain boundaries, cpTi significantly enhances cell adhesion [36]. A coating made up of a combination of cpTi and hydroxyapatite (Ca_5_(PO_4_)_3_OH) resulted in an HA coating thickness of 90 ± 30 µm. Traces of heavy metal elements were below 50 ppm, the mass fraction of the crystalline phases of HA was higher than 50% and its crystallinity value was above 45%. The porosity of the coating was ≤30%, the roughness depth Rt was 50 ± 20 µm (according to ISO 4287 [37] and 4288 [38]), the roughness Ra was 8-3 (ISO 4287, 4288), the tensile strength was ≥15 MPa (ISO 13779 [39], ASTM F 1147 [40]), and the shear strength was ≥20 (ASTM F 1044 [41]). Studies have shown that HA coatings on metallic prostheses lead to a shortening of the healing process [42]. The cpTi +HA coating on metallic prostheses combines the positive osseointegrative properties of both materials [43]. The combination of cpTi with tricalcium phosphate (Bonit^®^) led to a deposited layer of 20 ± 10 µm thickness and to a tensile strength of ≥15 MPa (ISO 13779-2). The chemical composition was built of the two phases, 70% Brushite (CaHPO_4_·2 H_2_O) and 30% hydroxyapatite. Due to its mineral components and bioactivity being significantly similar to that of human bone, this material promotes osseointegration and reduces the time taken for the implant to attach to the bone. For these reasons, hydroxyapatite and tricalcium phosphate are among the most commonly used calcium phosphates in biomedicine [44]. Rough-blasted TiAl6V4 was produced to generate a surface structure with micrometer-sized unevenness in which the osteoblasts could easily attach themselves. This led to the best possible conditions for the anchoring of osteoblasts, osseointegration, and a stable connection between bone and prosthesis without coatings [45]. All materials discussed in this study were sterilized using gamma irradiation according to standardized protocols.

**Bacterial strains and culture:***S. aureus* Newman (ATCC 25904, Wesel, Germany) and *S. epidermidis* (ATCC 14990, Wesel, Germany) were inoculated in LB broth containing 10 g/L tryptone, 5 g/L yeast extract (both Carl Roth), and 5 g/L sodium chloride (Merck, Darmstadt, Germany). For each experimental run, one overnight culture was prepared per strain using one CRYOBANK^®^ pearl (MAST Group, Reinfeld, Germany). Cultures were incubated at 37 °C and 90 rpm.

**Biofilm growth assay:** Bacterial cells from the overnight culture were inoculated on the discs with different surface modifications in 24-well untreated clear polystyrene plates (Corning^®^, Wiesbaden, Germany). Starting cell numbers were adjusted to 1.5 × 10^8^ CFU/mL in LB broth. Into each well, 1.5 mL of the cell solution was pipetted, and the plate was sealed using the Breathe Easy^®^ sealing membrane (Merck, Darmstadt, Germany). Biofilms were grown at 37 °C and 90 rpm for 48 h. In each of the 21 experimental runs, a total of four discs per material and species were used. Additionally, a growth control without material discs was prepared, as well as a sterile control with an uninoculated medium.

**Biofilm collection:** Biofilms were collected under sterile conditions, and two of the four discs were pooled in one sample tube, resulting in two samples per material and experimental run. The medium was removed by gently pipetting it out of the well in order to leave the biofilm intact. Then, 1.5 mL phosphate buffered solution (PBS) was introduced into each well and the discs were lifted using tweezers. The biofilm was scraped off into the well using a mini cell scraper (Biotium, Freemont, CA, USA). The biofilm solution was transferred to a 3.5 mL tube and vortexed until no visible flakes were left. In contrast to the samples, the blanks were re-suspended in the medium and then transferred into 3.5 mL tubes. The samples (3 mL per pool) were then aliquoted for the corresponding tests.

**Live dead assay:** The ‘LIVE/DEAD^®^ BacLightTM Viability Kit (Invitrogen, Carlsbad, CA, USA) for microscopy and quantitative assays’ was used for flow cytometric cell enumeration and live–dead evaluation of the samples. A separate dye mix of Syto9^®^ and propidium iodide was prepared for each batch in a ratio of 1:1, and 1 mL of each sample was stained using 1 µL of the dye mixture. Incubation in the dark for 15 min was carried out at room temperature. A Cyflow^®^ Cube 6 flow cytometer (Sysmex Europe GmbH, Norderstedt, Germany) was used to analyze the samples. Subsequently, 100 µL of each sample was analyzed at a flow rate of 2 µL per second using a 488 nm laser, and measurements were performed twice per pool. A cleaning step was inserted between each measurement to avoid signal carryover.

**Protein quantification:** A ‘PierceTM BCA Protein Assay Kit’ (Thermo Fisher Scientific, Waltham, MA, USA) was used to measure the total protein content according to the manufacturer’s guidelines using a Multiskan Sky Microplate Spectrophotometer (Thermo Fisher Scientific) and an uncoated U-bottom 96-well (BRAND^®^, Sigma-Aldrich, Darmstadt, Germany) plate to read absorbance of the samples at 562 nm. Each pool was measured in duplicates. Results were compared to a standard curve of bovine serum albumin (BSA, supplied with the BCA kit; 0–2000 µg/mL, Thermo Fisher Scientific, Waltham, MA, USA).

**Polysaccharide quantification:** An adapted version of Cuesta et al.’s (2003) [46] sulfuric acid phenol method was used: 250 µL of each sample, growth control, and blank were transferred into a heat-resistant glass tube, and 250 µL 99.5% phenol and 750 µL 95–98% sulfuric acid were added to each tube. The sample tubes were then sealed with aluminum foil, ensuring air circulation, and were vortexed for at least 20 s. The samples were incubated in a water bath for 10 min at 100 °C and their temperatures reduced to 25 °C in a separate water bath. Samples were vortexed and then 250 µL was pipetted into 96-well plates and measured at 490 nm in the Multiskan Sky Microplate Spectrophotometer using an uncoated U-bottom 96-well plate. Samples were measured in duplicates and compared to a standard glucose curve (Merck; 0–1.5 µg/mL).

**Statistical analyses:** Statistical analyses of flow cytometry, protein, and polysaccharide data were performed in SPSS (IBM, version 25) [47]. Data were tested for normal distribution using the Kolmogorov–Smirnov test with Lilliefors correction and were found to be non-Gaussian. Therefore, the Kruskal–Wallis one-way ANOVA was used for determining statistical differences in a pairwise comparison format.

**RNA isolation:** The Monarch^®^ Total RNA Miniprep Kit (New England BioLabs, Ipswich, MA, USA) was used to extract the total RNA of the samples of three independent experimental runs. The isolation was carried out following the manufacturer’s guidelines for tough-to-lyse samples (enzymatic approach) and was adapted at the enzymatic lysis step, utilizing 3 mg/mL lysozyme (provided with the kit) and additionally 0.1 mg/mL lysostaphin (Sigma-Aldrich, Darmstadt, Germany) for incubation at 37 °C for 25 min at 350 rpm.

**RT Q-PCR:** A blend of oligo (dT) and random hexamer primers was used to reverse-transcribe 1 μg RNA using the iScript cDNA Synthesis Kit (BioRad Laboratories Inc., Veenendal, The Netherlands). The SsoAdvanced Universal SYBR Green Supermix was used to amplify the samples which were then measured using the CFX96 Touch (BioRad Laboratories Inc.), as described elsewhere [48]. Each qPCR run consisted of a standard 3-step PCR temperature protocol with an annealing temperature of 60 °C followed by a melting curve protocol to confirm a single gene-specific peak and to detect primer dimerization. The ΔΔCt method was used to calculate the relative quantification of expression levels using the geometric mean of the internal controls TBP (TATA-box binding protein) and RPLP0 (lateral stalk ribosomal protein, subunit P0). The expression levels (Ct) of the target genes were normalized to the reference genes (ΔCt) and the ΔΔCt value was calculated using the difference between the ΔCt value of the test sample and the ΔCt of the control sample. The final expression ratio was expressed as 2ΔΔCt [49]. Primers used for RT-qPCR (Eurofins Genomics, Ebersberg, Germany) are listed in Table 1.

## 3. Results

### 3.1. Surface Characteristics

Scanning electron microscopy (SEM, FEI Quanta 250 FEG, Thermo Fisher Scientific, Hillsboro, OR, USA) and energy-dispersive X-ray spectroscopy (EDX, Octane Elect Plus Silicon Drift Detector by EDAX Ametek, NJ, USA) were used to analyze the different surface modifications. Figure 1 shows microphotographs in 1000× and 5000× magnifications for each surface. The surfaces could be grouped visually into smooth (TiAl6V4, TiN, Ag) and rough (rough-blasted TiAl6V4, cpTi, cpTi + TCP) with TiAl6V4 + cpTi + HA somewhere in between these groups (Figure 1F).

Energy-dispersive X-ray analysis results are depicted in Figure 2 for TiAl6V4, TiAl6V4 + Ag, TiAl6V4 + TiN, TiAl6V4 rough-blasted, and TiAl6V4 + cpTi.

### 3.2. Protein Assay

In each of the 21 independent experimental runs, two of the four samples per material were pooled, and from each pool proteins were measured in duplicate (*n* = 84). The untreated original alloy TiAl6V4 was used as a control sample for comparison. *S. aureus* biofilms were developed after 48 h of incubation, showing the lowest concentrations on the untreated alloy TiAl6V4 with a mean value of 142.6 µg/mL ± 30.8 µg/mL (minimum value 92.5 µg/mL, maximum value 211.3 µg/mL), followed by the ceramic coating TiN (153.1 ± 33.5 µg/mL) and Ag with 168.4 µg/mL ± 38.7 µg/mL. The highest levels of proteins in the biofilm were detected on TiAl6V4 + cpTi + TCP, with a mean value of 264.0 µg/mL ± 68.1 µg/mL (min. 148.9 µg/mL, max. 466.8 µg/mL). The second highest protein levels of 253.6 µg/mL ± 81.8 µg/mL were detected on the coating combination of commercially pure titanium (cpTi) with HA. An overview of the data for *S. aureus* is shown in Figure 3A as the mean ± SD. Statistically significant differences in the protein values of the *S. aureus* biofilms developed on TiAl6V4 and the surface modified materials have been found for all modifications, with the exception of TiN (Figure 3A). For *S. epidermidis*, an overview of the mean ± SD is given in Figure 3B. Generally, the values were in a lower range (62.7 µg/mL (TiAl6V4)–411.4 µg/mL (cpTi)) compared to *S. aureus* (92.5 µg/mL (TiAl6V4)–561.2 µg/mL (HA)). The TiN-coated surface was the only modification that reduced the total protein concentrations in the *S. epidermidis* biofilms with a mean protein value of 120.4 µg/mL ± 36.5 µg/mL (min. 69.5 µg/mL, max. 228.7 µg/mL) when compared to the control TiAl6V4 (125.1 µg/mL ± 35.8 µg/mL). This was followed by the Ag-coated surface (142.9 µg/mL ± 32.7 µg/mL), which resulted in a higher protein value than the control. The highest protein values were again seen on the TCP-modified surface (237.2 µg/mL ± 52.9 µg/mL) followed by the cpTi surface, showing a mean of 224.3 µg/mL ± 72.4 µg/mL. Only the rough materials resulted in a protein value for the *S. epidermidis* biofilms that differed to a statistically significant extent from those developed on TiAl6V4 (Figure 3B).

### 3.3. Polysaccharide Assay

Polysaccharide levels of biofilms are often measured by breaking down polysaccharides into glucose and subsequently determining the glucose concentrations [46]. This technique was used in these experiments, and the values of 21 independent experimental runs are summarized in Figure 4A for *S. aureus* and in Figure 4B for *S. epidermidis* biofilms. For each material, four samples were produced and mixed into two pools (*n* = 84). From those pools, duplicate measurements were performed, and the resultant mean ± SD values are shown compared to the untreated TiAl6V4 alloy, which was used as a control. The overall values of the two strains were comparable in their ranges, with the values for *S. aureus* lying between a minimum value of 0.333 µg/mL (TiAl6V4) and a maximum value of 1.984 µg/mL (cpTi), and for *S. epidermidis* between 0.331 µg/mL (TiAl6V4 and Ag) and 6.114 µg/mL (TCP). For all alloys, *S. aureus* biofilms showed an increase in glucose production when compared to the control material TiAl6V4 (mean value 0.465 µg/mL ± 0.075 µg/mL, min. 0. 333 µg/mL, max. 0. 672 µg/mL). TiN displayed the smallest increase in glucose levels (0.472 µg/mL ± 0.068 µg/mL), followed by Ag (0.489 µg/mL ± 0.084 µg/mL). The highest concentration was detected on the cpTi surface (0.699 µg/mL ± 0.277 µg/mL), followed by TCP (0.636 µg/mL ± 0.183 µg/mL). Statistical differences in the glucose concentrations of *S. aureus* biofilms were detected between TiAl6V4 and cpTi, as well as TCP (Figure 4A). *S. epidermidis* biofilms on TiN (0.465 µg/mL ± 0.081 µg/mL) and Ag (486 µg/mL ± 0.097 µg/mL) showed a decrease in glucose levels compared to the control material TiAl6V4 (mean value 0.514 µg/mL ± 0.212 µg/mL, min. 0.333 µg/mL, max. 1.950 µg/mL). All other materials induced a higher glucose level in biofilm formation. The smallest increase was found on the rough-blasted alloy (rb), with a mean of 0.531 µg/mL ± 0.129 µg/mL. The highest glucose levels were detected on TCP followed by HA, with 0.773 µg/mL ± 0.819 µg/mL and 0.735 µg/mL ± 0.341 µg/mL, respectively. Statistically the surface modifications cpTi and TCP were different from the base material (Figure 4B).

### 3.4. Live/Dead Assay

For an assessment of bacterial cell vitality, a live/dead assay was performed. In this measurement, intact cells showed green fluorescence, dead cells were fluorescing red, and damaged cells showed orange fluorescence. The test was calibrated using known proportions of live and dead cells. Measurements were performed for 21 independent experimental runs, and again, two samples per material were mixed, resulting in two pools per material and experimental run (*n* = 84). Flow cytometry was performed twice for every pool; no dead cells could be detected for any material and bacterial strain. The unmodified TiAl6V4 alloy was again used as a control and for comparison. No significant difference was seen in the cell counts for any of the surfaces (Figure 5).

### 3.5. RT-qPCR

Biofilm-associated genes of S. *aureus* and *S. epidermidis* (Table 1) were quantified using an RT-qPCR assay. The intercellular adhesion group (*ica*) of biofilm-associated genes was represented with *icaA* and *icaC* for *S. aureus*. As shown in Figure 6A, *icaA* was expressed significantly less (according to a 95% significance level in the *t*-test) when analyzing biofilms grown on TiN, Ag and HA. For *icaC*, a significantly lower value was detected on TiN and TCP (Figure 6B). Fibronectin binding gene A (*fnbA*) was significantly less expressed on TiN, TCP, and HA, whereas *fnbB* expression was significantly lower in TiN, Ag, cpTi, and TCP. Bacterial ligand clumping factor B (clfB) showed increased expression for TCP and HA, and the major autolysin (*atl*) for TiN and Ag. None of the *S. aureus* genes were significantly more expressed when compared to the base material. For *S. epidermidis*, however, significant increases in gene expression differences were found compared to the base material. *Atle* (the major autolysin of *S. epidermidis*) was significantly more expressed on rb, *aap* (the accumulation associated protein), and cpTi.

## 4. Discussion

Although *S. aureus* and *S. epidermidis* are some of the most intensively studied species when it comes to biofilm development, our understanding of how they interact with different surfaces is still only partial. Those interactions are of great interest, particularly in the field of orthopedics, and this study can provide more insight into the biofilm-influencing abilities of the investigated materials.

Proteins are among the most important molecules in the human body and have an important role to fulfil regarding implant materials because of their ability to adsorb quickly onto prostheses and thereby influence the biocompatibility of a foreign material. Protein binding depends on the surface charge of the protein and the prosthesis material, the surface structure of the protein and the prosthesis, and the size of the proteins [53].

Prosthesis materials can be modified by subsurface micro-structures (as in cpTi, rough-blasted TiAl6V4, TCP, and HA) to facilitate the binding of proteins. These structures can enhance the adherence of autologous proteins and osteoblasts, thereby promoting osseointegration. It is, however, crucial to consider that protein adhesion does not have exclusively positive effects; indeed, our protein assay showed that it also enables the rapid binding of bacteria. Proteins perform a key role in biofilm formation, maintenance, and expansion. Bacterial proteins, such as cell surface adhesins, membrane components, and extracellular proteins (including extracellular vesicles), are major contributors to the adhesion and migration of the biofilm on the surface of the prosthesis [28,54]. Furthermore, the extracellular matrix which is produced protects the bacteria from the human immune system and antibiotics and other substances [28,55,56]. In this sense, smooth surfaces, such as those resulting from TiAl6V4, TiN, and silver alloys on prostheses, can, to a certain extent, inhibit the adhesion of proteins. By reducing the binding of the corresponding proteins, bacterial colonization on an implant can be reduced or prevented, thus augmenting its antibacterial and antibiofilm properties.

cpTi, TCP, and HA surfaces showed increased biofilm protein content compared to the base TiAl6V4 material for both bacterial types tested, while Ag and rb showed an increased protein content only in the *S. aureus* tests. Based on the protein results, no treatment reduced the rate of creation of bacterial generated biofilm compared to that seen using the base TiAl6V4 material, and only TiN did not increase the rate of creation of such biofilms.

These results are consistent with the glucose measurements, which appear to be a less sensitive measure of differences in biofilm growth between different surfaces, but nevertheless show increased biofilm growth with both bacteria with the cpTi and TCP surfaces. As explained by Limoli et al., 2015, polysaccharides play a major role in the adhesion, protection, and structure of the biofilm [57]. In particular, polysaccharides, induced by the expression of the *ica operon*, are important players in biofilm development and are discussed further in the gene expression section below.

Generally, the rougher surfaces (rough-blasted TiAl6V4, cpTi, cpTi + TCP) showed a higher tendency for the bacterial cells to form more proteins and polysaccharides in their matrix compared to the surfaces appearing smoother (TiAl6V4, TiN, Ag) in the SEM pictures (see also Figure 1 and Table 2). A study by Palka et al. (2020) [26] also conclusively showed that rougher surfaces promote the susceptibility to microbial adhesion, amongst other bacterial species as well as for *S. aureus* and *S. epidermidis*. Similar findings, but for different base materials (Ti6Al4V, CoCr alloy and zirconium), have been reported by Minkiewicz-Zochniak et al., 2021 [24].

The cell count determination by means of flow cytometry was carried out to answer two questions. Firstly, to show whether the materials influence the proliferation of cells and, secondly, whether the investigated alloys have a cytotoxic effect. The flow cytometry revealed virtually identical results for both strains of bacteria for all alloys. Although the alloy platelets all had the same shape, the textures of their surfaces varied substantially, which had an impact on the strength of the biofilm, confirming previous findings by Stewart et al., 2017 [58]. Their research explains that while more complex surfaces encourage the production of proteins and polysaccharides for the extracellular matrix, they do not directly affect the proliferation of cells. This is consistent with our observation that while the rough alloys studied as part of this project manifested higher concentrations of proteins and polysaccharides (i.e., a notably pronounced extracellular matrix) compared to the smooth alloys, all textures yielded similar cell counts. The cell count measurements did not identify any differences in the cytotoxic properties of the alloys tested for either *S. aureus* or *S. epidermidis.* Additional experiments with longer incubation times and additional genetic analysis of the bacteria could help shed light on this specific issue.

When comparing the results of the genetic analysis to the other results, they seem at first glance to be contradictory (Table 2); while the surface modification only resulted in increased biofilm growth according to the polysaccharide and protein results, the only statistically significant changes in gene expression were reductions. However, when compared to earlier studies, the results support previous findings on gene regulation, as described in detail below. The gene expression was measured at the endpoint of each experimental series, resulting in values for 48 h biofilms, always compared to the base material at the same 48 h timepoint. Although we always detected an expression of the genes, the expression level varied on the different types of alloys. It is known that the *ica* family of genes is involved at the start of biofilm formation [50,59,60], particularly the formation of PIA (polysaccharide intercellular adhesin), an extracellular polysaccharide [61]. Resch et al., 2005, showed in their study that these genes are especially upregulated between 6 and 8 h after the start of biofilm development. Atshan et al. (2013), however, showed a slight upregulation for most of the *ica* operons after 24 h and none at 12 h or 48 h [50,60]. Our results for *icaA* and *icaC* are consistent with these findings, in that the *icaA* and *icaC* genes in the investigated strains were no longer active after 48 h of biofilm formation. TiN seemed to have the most direct influence on the expression level of these genes when compared to the base material—this might be an interesting anchor point for future studies. It is thus quite straightforward to connect the *ica* expression levels with the protein and polysaccharide measurements: those substances which build up during the initial phase of the biofilm development stay in the matrix, even when the genes involved in the initial process are downregulated.

The two fibronectin binding proteins, *fnbA* and *fnbB*, have been reported to contribute to biofilm formation in *S. aureus* [61,62]. Moreover, Atshan et al., 2013, also reported the peak of expression at 24 h of biofilm development, with a distinct decline after 48 h [50]. Our results showed decreased values compared to the base alloy after 48 h of biofilm development. This could be due to a different developmental pattern of our strains, resulting in an earlier total decline of the *fnb* gene expression on the different alloy types, again with TiN being the alloy with the most distinct influence on the gene expression levels.

The *clfB* (bacterial ligand clumping factor) gene, important for the interaction with the host extracellular ligands, also showed decreased values compared to the base alloy after 48 h (similarly to *fnbA* and *fnbB)*. This is consistent with the literature [50]. *Atl*, the major autolysine gene of *S. aureus* (corresponding to *atle* in *S. epidermidis*), is known to be important for the initial attachment [51,52,62], and therefore should no longer be upregulated after 48 h. This was true for all our measurement points of *S. aureus*. However, *S. epidermidis atle* on cpTi was still significantly higher compared to the base material values. Interestingly, all *atle* values of *S. epidermidis* showed a tendency to be increased, even after 48 h. This is in the line of the conclusions drawn by Patel et al., 2012, who found increased *atle* levels even after 48 h of biofilm development [52]. It would be interesting to test this with different *S. epidermidis* strains, or on different materials, to confirm the prolonged upregulation of this particular gene which is involved in initial attachment.

Finally, the *aap* (accumulation association protein) is common in the clinically relevant *S. epidermidis* biofilm building strains (even those that are *ica* negative) [59]. According to Patel et al., 2012, *aap* shows the fastest increase between 12 h and 24 h, but continues to increase between 24 and 48 h of biofilm development, supporting the findings of this study. Patel et al. (2012) [52] analyzed an additional gene (the *agr* quorum-sensing system), which also had the least variation in gene expression as a function of the different surface types at 48 h. This might explain why there was very little variation between *S. epidermidis aap* expression on surfaces in this study.

## 5. Conclusions

To the best of our knowledge, this study was the first to extensively study the connection of coatings with silver (Ag), titanium nitride (TiN), pure titanium (cpTi), combinations of cpTi and hydroxyapatite (HA), combinations of cpTi and tricalcium phosphate (TCP), and a rough-blasted surface of TiAl6V4 on the phenotypic and genotypic biofilm development of two bacterial species (*S. aureus* and *S. epidermidis*) often found in implant infections. Our findings lead to the conclusion that artificial implant surfaces have contradictory requirements; they must present a biocompatible surface to facilitate their integration into the host biological tissue, while at the same time not provide a conducive surface for bacterial-based biofilms to grow. This study considered the degree to which different surface alloy types affected the development of staphylococcal biofilms on their surfaces. Protein and polysaccharide measurements of the biofilms indicated that the comparatively smooth surfaces of the base TiAl6V4 alloy and titanium nitride (TiN) were the most effective at preventing biofilm growth, presumably because they prevented the organisms from attaching easily and therefore were best at suppressing bacterial infections. Interestingly, rough-blasting the TiAl6V4 surface to make the surface more biocompatible with the host tissue turned out to be the best option for creating a rough surface, because all the other coated surfaces seemed to provide a more conducive surface for the growth of bacterial biofilms with a higher protein and polysaccharide content. The fact that this was not associated with a greater cell count indicated a thicker extracellular matrix which protects the bacterial cells and makes them less likely to be penetrated by antibiotic substances [63].

## Figures and Tables

**Figure 1 jfb-12-00036-f001:**
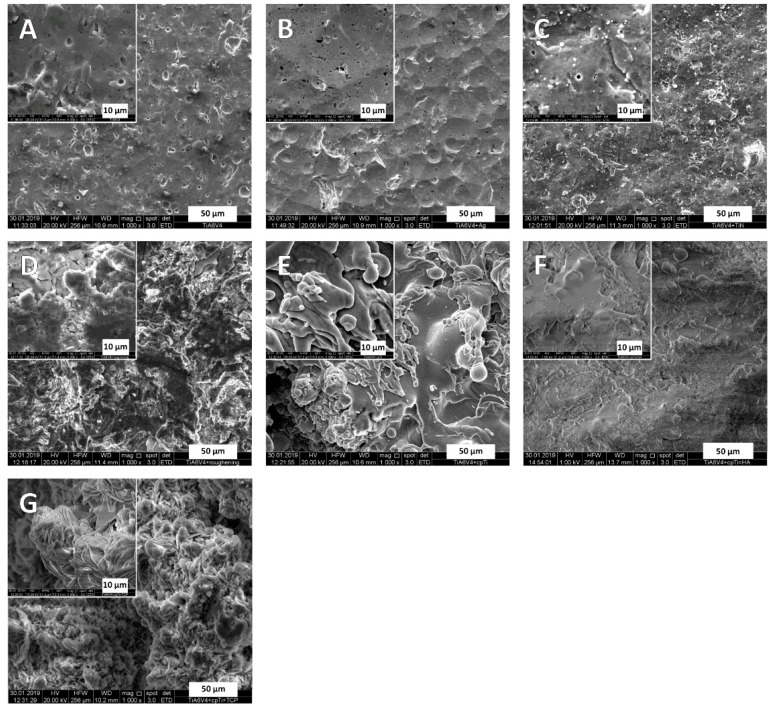
Scanning electron microphotographs of TiAl6V4 alloy and different surface modifications showing the corresponding textures. The background images were recorded with a magnification factor of 1000, and the inlays with a factor of 5000. The subfigures show the different surfaces: (**A**) TiAl6V4; (**B**) TiAl6V4 + Ag; (**C**) TiAl6V4 + TiN; (**D**) TiAl6V4 rough-blasted; (**E**) TiAl6V4 + cpTi; (**F**) TiAl6V4 + cpTi + HA; (**G**) TiAl6V4 + cpTi + TCP.

**Figure 2 jfb-12-00036-f002:**
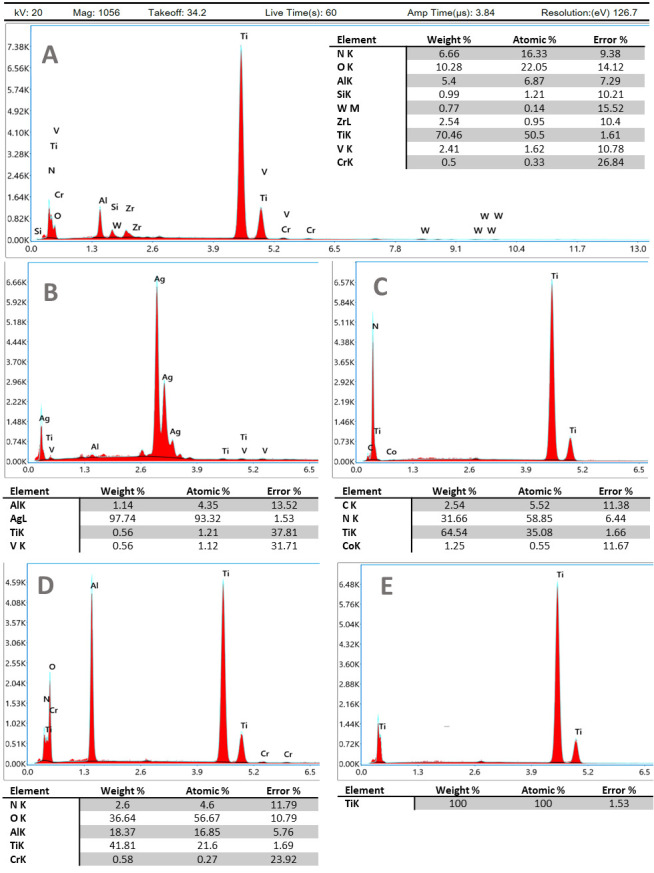
Energy-dispersive X-ray analysis of the surfaces investigated. Instrument values given in (**A**) also apply to (**B**–**E**). (**A**) TiAl6V4; (**B**) TiAl6V4 + Ag; (**C**) TiAl6V4 + TiN; (**D**) TiAl6V4 rough-blasted; (**E**) TiAl6V4 + cpTi.

**Figure 3 jfb-12-00036-f003:**
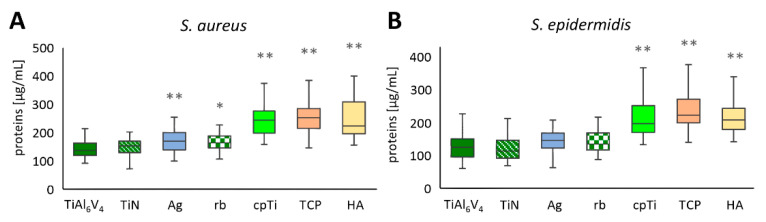
Total protein concentrations of the biofilms for *S. aureus* (**A**) and *S. epidermidis* (**B**). Differences between the base material TiAl6V4 and the surface modification types have been calculated, and statistically significant differences are marked as * (adjusted significance < 0.05 according to the Bonferroni correction for the Kruskal–Wallis test) or ** (adj. sig. ≤ 0.01).

**Figure 4 jfb-12-00036-f004:**
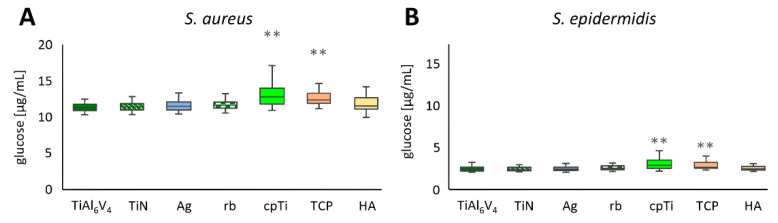
Glucose concentration of *S. aureus* (**A**) and *S. epidermidis* (**B**) biofilms on the different titanium-alloy surface modifications. Differences between the base material TiAl6V4 and the surface modification types have been calculated and statistically significant differences are marked as ** (adjusted significance < 0.01 according to the Bonferroni correction for the Kruskal–Wallis test).

**Figure 5 jfb-12-00036-f005:**
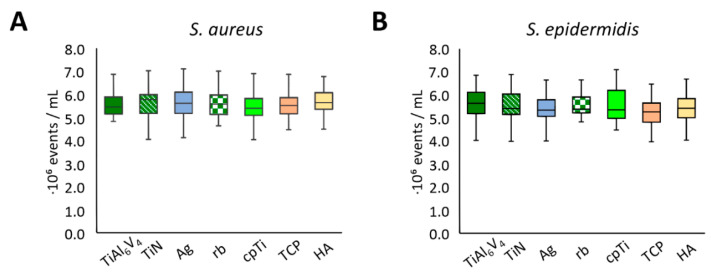
Flow cytometry live cell counts for *S. aureus* (**A**) and *S. epidermidis* (**B**).

**Figure 6 jfb-12-00036-f006:**
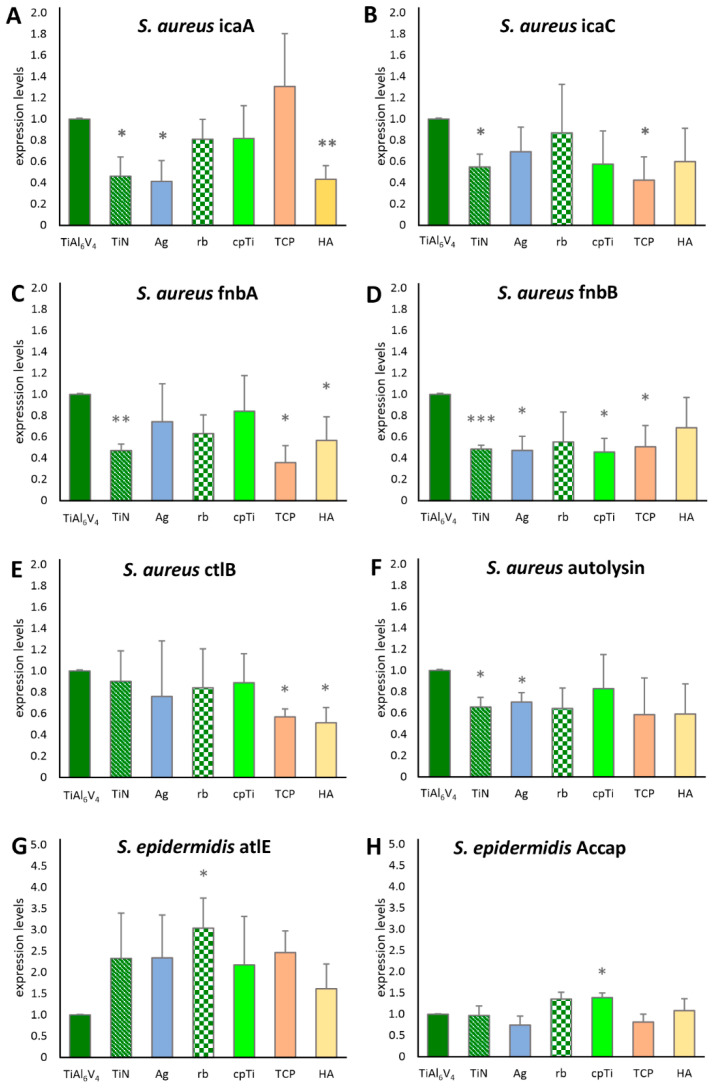
Expression levels of biofilm-associated genes for *S. aureus* (**A**–**F**) and *S. epidermidis* (**G**,**H**). Statistically significant differences between the base material TiAl6V4 and the surface modification types are marked as * (*p* < 0.05), ** (*p* < 0.01) and *** (*p* < 0.001).

**Table 1 jfb-12-00036-t001:** Primers used for RT-qPCR.

Strain	Gene	Primer Forward	Primer Reverse
*S. aureus*	*icaA* [50]	5-GAGGTAAAGCCAACGCACTC-3	5-CCTGTAACCGCACCAAGTTT-3
-	*icaC* [50]	5-CTTGGGTATTTGCACGCATT-3	5-GCAATATCATGCCGACACCT-3
-	*fnbA* [50]	5-AAATTGGGAGCAGCATCAGT-3	5-GCAGCTGAATTCCCATTTTC-3
-	*fnbB* [50]	5-ACGCTCAAGGCGACGGCAAAG-3	5-ACCTTCTGCATGACCTTCTGCACCT-3
-	*clfB* [50]	5-AACTCCAGGGCCGCCGGTTG-3	5-CCTGAGTCGCTGTCTGAGCCTGAG-3
-	*atl* [51]	5-TTTGGTTTCCAGAGCCAGAC-3	5-TTGGGTTAAAGAAGGCGATG-3
*S. epidermidis*	*atle* [52]	5-TGTCCTGCTTTCACGTATGA-3	3-TCTTTGGAATTGGTGCATTT-5
-	*aap* [52]	5-TGATCGGATCTCCATCAACT-3	3-AAGGTAGCCAAGAGGACGTT-5

**Table 2 jfb-12-00036-t002:** Comparison of statistically significant differences between the TiAl6V4 protein, glucose, and gene expression levels of *S. aureus* and *S. epidermidis*. ↑ indicates increased levels, ↓ indicates decreased levels. **↑** or **↓** (*p* < 0.05), **↑↑** or **↓↓** (*p* < 0.01) and **↑↑↑** or **↓↓↓** (*p* < 0.001).

Alloys Compared to TiAl6V4	*S. aureus*								*S. epidermidis*			
	Proteins	Glucose	*icaA*	*icaC*	*fnbA*	*fnbB*	*clfB*	*atl*	Proteins	Glucose	*atle*	*aap*
TiN	-	-	↓	↓	↓↓	↓↓↓		↓	-	-	-	-
Ag	↑↑	-	↓	-	-	↓		↓	-	-	-	-
rb	↑	-	-	-	-	-	-	-	-	-	↑	-
cpTi	↑↑	↑↑	-	-	-	↓	-	-	↑↑	↑↑	-	↑
TCP	↑↑	↑↑	-	↓	↓	↓	↓	-	↑↑	↑↑	-	-
HA	↑↑	-	↓	-	↓		↓	-	↑↑	-	-	-

## Data Availability

The data that support the findings of this study are available from the corresponding author upon reasonable request.

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
