# Peer review of "TiAl6V4 Alloy Surface Modifications and Their Impact on Biofilm Development of S. aureus and S. epidermidis"

_jfb, 2021, doi:10.3390/jfb12020036_

Round 1
Reviewer 1 Report
The paper is interesting for the advanced experimental approach even if the topic is already well know.
Some minor corrections are requested.
Abstract:
1) "Various types of titanium-aluminum-vanadium (TiAl6V4) alloy surface modifications were tested ...": it is not clear which is the set of investigated surfaces and because of this, the results summarized later in the abstract are not clear. Make in the abstract a short list of the types of investigated surfaces.
2) "whereas rough surfaces are most favorable when it comes to bacterial infections since they prevent an easy attachment of pathogenic organisms": it is not clear when the base alloy/TiN coating or a rough surface is favourable.
Materials and methods:
1) "The silver coated TiAl6V4 discs were produced showing an electrolytically deposited": why did you select this coating? is there any commercial product like this? I am a bit surprised because of this selection. It is well known that a thick silver metal coating is cytotoxic. Some coatings were investigated in the literaure with a very low amount of silver (Nanomaterials 2020, 10(1), 120; https://doi.org/10.3390/nano10010120) in order to get a balance between antibacterial action and cytocompatibility. Can you give a motovation of your material selection?
Results
1) A part from SEM pictures, topographical measurements of roughness must be reported (Ra or Sa ...) in order to make a quantification of the differences among the tested surfaces
Author Response
Dear reviewer,
We thank you for the comments and suggestions. Please find below our corrections and/or answers to the points you raised (italics):
Abstract:
1) "Various types of titanium-aluminum-vanadium (TiAl6V4) alloy surface modifications were tested ...": it is not clear which is the set of investigated surfaces and because of this, the results summarized later in the abstract are not clear. Make in the abstract a short list of the types of investigated surfaces.
A list of the different surface modifications has been added to the abstract: ´…Various types of titanium-aluminum-vanadium (TiAl6V4) alloy surface modifications (coatings with silver (Ag), titanium nitride (TiN), pure titanium (cpTi), combination of cpTi and hydroxy-apatite (HA), combination of cpTi and tricalcium phosphate (TCP) and a rough blasted surface of TiAl6V4) have been investigated to assess their effects on biofilm development. …´
2) "whereas rough surfaces are most favorable when it comes to bacterial infections since they prevent an easy attachment of pathogenic organisms": it is not clear when the base alloy/TiN coating or a rough surface is favourable.
This mistake has been clarified in this manner: ´… According to the findings of this study the base alloy TiAl6V4 and the TiN coated surface are the most promising materials for biofilm reduction. Rough surfaces are most favorable when it comes to bacterial infections since they allow an easy attachment of pathogenic organisms. …´
Materials and methods:
1) "The silver coated TiAl6V4 discs were produced showing an electrolytically deposited": why did you select this coating? is there any commercial product like this? I am a bit surprised because of this selection. It is well known that a thick silver metal coating is cytotoxic. Some coatings were investigated in the literaure with a very low amount of silver (Nanomaterials 2020, 10(1),120; https://doi.org/10.3390/nano10010120) in order to get a balance between antibacterial action and cytocompatibility. Can you give a motovation of your material selection?
We agree with the reviewer in the general conclusions on the influence of silver coatings on cell cytotoxicity. The topic is very controvers and there are studies showing high silver levels in blood and serum of patients and other only reporting a minor influence. However silver coatings also show promising results in fighting bacterial infections. In the light of this controversy, and as the study is focusing on the influence of the coatings on the bacterial adhesion processes, we wanted to include a solid silver coating in the study in order to add to the knowledge base for future studies.
In the material and methods section the following has been added to address this controversy shortly: ´… However, this cytotoxicity can also affect bone and tissue cells and release of silver ions might have a toxic effect on the patients (high serum and blood Ag levels) as well [31,32].….´
Two references (Wyatt et al. 2019 and Lohberger et al. 2021) have been added as well to support this topic.
Results
A part from SEM pictures, topographical measurements of roughness must be reported (Ra or Sa ...) in order to make a quantification of the differences among the tested surfaces
The appropriate units have been added to the values (as far as they were available from the manufacturer) given in the material and methods section as follows:
´… The TiN-target used for deposition had the following elemental composition (in %): Fe £ 0.2, O £ 0.25, C £ 0.1, N £0.03, H £ 0.0125 and the remainder consisting of titanium. The coating had a thickness of 5.5±1.5 µm, an adhesive tensile strength of ³22 MPa and a layer roughness of Ra £ 0.05 µm. …´
´… The titanium coating was applied using a vacuum plasma spray (VPS) of commercially pure titanium resulting in a 300 ± 50 µm coating thickness and an adhesive tensile strength of greater than 22 MPa and an average roughness of Ra 50 ± 15 µm. … ´
For HA: ´… The porosity of the coating was £ 30%, the roughness depth Rt was 50 ± 20 µm (according to ISO 4287 and 4288), the roughness Ra was 8-3 (ISO 4287, 4288) the tensile strength was ³ 15 MPa (ISO13779, ASTM F 1147) and the shear strength was ³ 20 (ASTM F 1044). …´
Reviewer 2 Report
The authors have written a well-structured and thought research article. I just have a few minor suggestion: please add more recent literature regarding the topic of biofilm formation and titanium surfaces- I'd suggest adding the following in the discussion section:
Minkiewicz-Zochniak A, Jarzynka S, Iwańska A, et al. Biofilm Formation on Dental Implant Biomaterials by Staphylococcus aureus Strains Isolated from Patients with Cystic Fibrosis. Materials (Basel). 2021;14(8):2030. Published 2021 Apr 17. doi:10.3390/ma14082030
Palka L, Mazurek-Popczyk J, Arkusz K, Baldy-Chudzik K. Susceptibility to biofilm formation on 3D-printed titanium fixation plates used in the mandible: a preliminary study. J Oral Microbiol. 2020;12(1):1838164. Published 2020 Oct 29. doi:10.1080/20002297.2020.1838164
I'd also suggest thinking if the roughness value shouldn't be given in 3D not 2D. There are also some language mistakes so I'd recommend editing the article before resubmission (e.g. line 18- change "were tested to assess their" to "have been tested"; line 24- are you sure the word "prevent" should be used there?; line 34 "whose main causing agents" into "caused mainly by"; and so on).
Author Response
Dear reviewer,
We thank you for the comments and suggestions. Please find below our corrections and/or answers to the points you raised (italics):
The authors have written a well-structured and thought research article. I just have a few minor suggestion: please add more recent literature regarding the topic of biofilm formation and titanium surfaces- I'd suggest adding the following in the discussion section:
Minkiewicz-Zochniak A, Jarzynka S, Iwańska A, et al. Biofilm Formation on Dental Implant Biomaterials by Staphylococcus aureus Strains Isolated from Patients with Cystic Fibrosis. Materials (Basel). 2021;14(8):2030. Published 2021 Apr 17. doi:10.3390/ma14082030
Palka L, Mazurek-Popczyk J, Arkusz K, Baldy-Chudzik K. Susceptibility to biofilm formation on 3D-printed titanium fixation plates used in the mandible: a preliminary study. J Oral Microbiol. 2020;12(1):1838164. Published 2020 Oct 29. doi:10.1080/20002297.2020.1838164
Thank you for the valued suggestions, both references have been added in the discussion as follows:
´… Generally, the rougher surfaces (rough blasted TiAl6V4, cpTi, cpTi + TCP) show a higher tendency for the bacterial cells to form more proteins and polysaccharides in their matrix compared to the surfaces appearing smoother (TiAl6V4, TiN, Ag) in the SEM-pictures (see also figure 1 and table 2). A study by Palka et al. (2020) [27] also showed conclusively that rougher surfaces promote the susceptibility to microbial adhesion, amongst other bacterial species also for S. aureus and S. epidermidis. Similar findings, but for different base materials (Ti6Al4V, CoCr alloy and zirconium) have been reported by Minkiewicz-Zochniak et al. (2021) [25]….´
I'd also suggest thinking if the roughness value shouldn't be given in 3D not 2D.
Where available from the manufacturer the roughness values are given in the material and methods section and now the appropriate units have been added:
´… The TiN-target used for deposition had the following elemental composition (in %): Fe ≤ 0.2, O ≤ 0.25, C ≤ 0.1, N ≤ 0.03, H ≤ 0.0125 and the remainder consisting of titanium. The coating had a thickness of 5.5±1.5 µm, an adhesive tensile strength of ≥ 22 MPa and a layer roughness of Ra ≤ 0.05 µm. …´
´… The titanium coating was applied using a vacuum plasma spray (VPS) of commercially pure titanium resulting in a 300 ± 50 µm coating thickness and an adhesive tensile strength of greater than 22 MPa and an average roughness of Ra 50 ± 15 µm. … ´
For HA: ´… The porosity of the coating was ≤ 30%, the roughness depth Rt was 50 ± 20 µm (according to ISO 4287 and 4288), the roughness Ra was 8-3 (ISO 4287, 4288) the tensile strength was ≥ 15 MPa (ISO13779, ASTM F 1147) and the shear strength was ≥ 20 (ASTM F 1044). …´
There are also some language mistakes so I'd recommend editing the article before resubmission
The requested changes have been made (see below) and an additional native English editor has read and revised the article.
e.g. line 18- change "were tested to assess their" to "have been tested"
reads now: ´… V Various types of titanium-aluminum-vanadium (TiAl6V4) alloy surface modifications (coatings with silver (Ag), titanium nitride (TiN), pure titanium (cpTi), combination of cpTi and hydroxy-apatite (HA), combination of cpTi and tricalcium phosphate (TCP) and a rough blasted surface of TiAl6V4) have been investigated to assess their effects on biofilm development. …´
line 24- are you sure the word "prevent" should be used there?
Thank you for the remark, this was a mistake and is corrected now: ´… According to the findings of this study the base alloy TiAl6V4 and the TiN coated surface are the most promising materials for biofilm reduction. Rough surfaces are most favorable when it comes to bacterial infections since they allow an easy attachment of pathogenic organisms. …´
line 34 "whose main causing agents" into "caused mainly by"
reads now: ´… Periprosthetic infections following the surgical treatment of bone defects are some of the most serious complication in orthopedics and are caused mainly by Staphylococcus aureus (S. aureus) and Staphylococcus epidermidis (S. epidermidis) [1]. … ´
Reviewer 3 Report
The present research is dealing with the surface modifications of TiAl6V4 alloy and the development of staphylococcal biofilms on their surfaces. I have the following comments and suggestion:
- The introduction part need to be extended with the most important findings, regarding to the coated TiAl6V4
- There are also several dozens of studies regarding to different coated/uncoated TiAl6V4 bacterial resistance, at least a paragraph need to be introduced to the introduction.
- Please emphasize the novelty of this research, both in introduction and conclusion part.
- The SEM images scale is barely visible, please magnify it.
- The quality of the figures 3, 4, 5 and 6 is poor, please replace them with better quality and colored figures it is possible.
Author Response
Dear reviewer,
We thank you for the comments and suggestions. Please find below our corrections and/or answers to the points you raised:
- The introduction part need to be extended with the most important findings, regarding to the coated TiAl6V4
- There are also several dozens of studies regarding to different coated/uncoated TiAl6V4 bacterial resistance, at least a paragraph need to be introduced to the introduction.
- Please emphasize the novelty of this research, both in introduction and conclusion part.
The following paragraph has been added to the introduction and the beginning of the next paragraph has been changed accordingly:
Numerous physical and chemical surface modifications can and have been applied to metallic biomaterials in general [11] and to titanium-based materials specifically [12] most recently also applying nano-modifications [13]. This led to the invention of many new types of titanium based orthopedic materials with not only a higher resistance to corrosion and better tribological properties [14,15], but also improved mechanical properties [16–19]. Also, biocompatibility [19]and resistance to bacterial infections [20–23] can be positively influenced by those newly developed materials. As mentioned above, especially S. aureus and S. epidermidis are prone to cause device related infections and they are used in many biofilm related studies in the field of orthopedics [6,21,22,24–28].
For this study, a pure TiAl6V4 base [5,28,29] was modified implementing silver (Ag), titanium nitride (TiN), pure titanium (cpTi), cpTi with hydroxyapatite (HA) and cpTi with tricalcium phosphate (TCP) surface coatings as well as rough blasting (rb) the surface of TiAl6V4. Using an in-vitro model, the influence of this newly developed surface modifications on S. aureus and S. epidermidis biofilm development has been studied using protein and glucose quantification assays, live/dead cell distinction flow cytometric measurements as well as the RT-qPCR of genes associated with biofilm development.
The conclusions now start with this sentence: ´… To the best of our knowledge this study was the first to extensively study the con-nection of coatings with silver (Ag), titanium nitride (TiN), pure titanium (cpTi), com-bination of cpTi and hydroxyapatite (HA), combination of cpTi and tricalcium phos-phate (TCP) and a rough blasted surface of TiAl6V4 on the phenotypic and genotypic biofilm development of two bacterial species (S. aureus and S. epidermidis) often found in implant infections. …´
4. The SEM images scale is barely visible, please magnify it.
The scale bars have been magnified.
5. The quality of the figures 3, 4, 5 and 6 is poor, please replace them with better quality and colored figures it is possible.
High resolution color figures have been included in the manuscript an can also be uploaded separately if requested by the editorial office.